# How Temperatures May Affect the Synthesis of Fatty Acids during Olive Fruit Ripening: Genes at Work in the Field

**DOI:** 10.3390/plants12010054

**Published:** 2022-12-22

**Authors:** Cibeles Contreras, Pierluigi Pierantozzi, Damián Maestri, Martín Tivani, Peter Searles, Magdalena Brizuela, Fabricio Fernández, Alejandro Toro, Carlos Puertas, Eduardo R. Trentacoste, Juan Kiessling, Roberto Mariotti, Luciana Baldoni, Soraya Mousavi, Paula Fernandez, Sebastián Moschen, Mariela Torres

**Affiliations:** 1Estación Experimental Agropecuaria San Juan, Instituto Nacional de Tecnología Agropecuaria (INTA), Consejo Nacional de Investigaciones Científicas y Técnicas (CONICET), San Juan 5427, Argentina; 2Instituto Multidisciplinario de Biología Vegetal, X5000 IMBIV—CONICET—Universidad Nacional de Córdoba, Córdoba 5000, Argentina; 3Centro Regional de Investigaciones Científicas y Transferencia Tecnológica de La Rioja, 5301 CRILAR La Rioja—UNLaR-SEGEMAR-UNCa, CONICET, Anillaco 5301, Argentina; 4Estación Experimental Agropecuaria Catamarca, INTA, Sumalao 4705, Argentina; 5Estación Experimental Agropecuaria Cerro Azul, INTA, Cerro Azul 3313, Argentina; 6Estación Experimental Agropecuaria Junín, INTA, Junín 5573, Argentina; 7Agencia de Extensión Rural Centenario, INTA, Plottier 8316, Argentina; 8CNR—Institute of Biosciences and Bioresources (IBBR), 06128 Perugia, Italy; 9Instituto de Agrobiotecnología y Biología Molecular (IABiMo—INTA-CONICET), Consejo Nacional de Investigaciones Científicas y Técnicas (CONICET), Centro de Investigaciones en Ciencias Agronómicas y Veterinarias, INTA, Hurlingham 1686, Argentina; 10Escuela de Ciencia y Tecnología, Universidad Nacional de San Martín, San Martín 1650, Argentina; 11Facultad de Ciencias Exactas y Naturales, Universidad de Buenos Aires, C1428EGA Ciudad Autónoma de Buenos Aires, Viamonte 2671, Argentina; 12Estación Experimental Agropecuaria Famaillá, INTA, CONICET, Famaillá 4132, Argentina

**Keywords:** *Olea europaea* L., fatty acid composition, gene expression, desaturase genes, thermal regime

## Abstract

A major concern for olive cultivation in many extra-Mediterranean regions is the adaptation of recently introduced cultivars to environmental conditions different from those prevailing in the original area, such as the Mediterranean basin. Some of these cultivars can easily adapt their physiological and biochemical parameters in new agro-environments, whereas others show unbalanced values of oleic acid content. The objective of this study was to evaluate the effects of the thermal regime during oil synthesis on the expression of fatty acid desaturase genes and on the unsaturated fatty acid contents at the field level. Two cultivars (Arbequina and Coratina) were included in the analysis over a wide latitudinal gradient in Argentina. The results suggest that the thermal regime exerts a regulatory effect at the transcriptional level on both *OeSAD2* and *OeFAD2-2* genes and that this regulation is cultivar-dependent. It was also observed that the accumulated thermal time affects gene expression and the contents of oleic and linoleic acids in cv. Arbequina more than in Coratina. The fatty acid composition of cv. Arbequina is more influenced by the temperature regime than Coratina, suggesting its greater plasticity. Overall, findings from this study may drive future strategies for olive spreading towards areas with different or extreme thermal regimes serve as guidance for the evaluation olive varietal patrimony.

## 1. Introduction

In the last two decades, the increasing demand for olive oil and table olives has led to the expansion of olive cultivation from the traditional Mediterranean area towards some countries in the southern hemisphere, notably Argentina, Chile, Peru and Australia. In most of these countries, olive growing takes place in regions having rainfall and thermal regimes which differ greatly from those of the Mediterranean countries [1]. As a consequence, unexpected reproductive, physiological and biochemical responses have been reported for some cultivars growing in Argentina and Australia [1,2,3,4,5,6,7].

In Argentina, olive cultivation has been developed mainly in the northwestern and central–western regions, in valleys bordering the Andes mountains. These regions are characterized by arid or semiarid conditions, with annual rainfall generally not exceeding 200 mm and average winter and spring temperatures significantly higher than those typical of the Mediterranean basin [1].

Whereas numerous studies have made it possible to identify genes involved in the synthesis of the main fatty acids (FA) of olive oil [8,9,10,11,12,13,14,15,16], increasing evidence shows significant genotype *x* environment interactions with respect to fatty acid composition in many olive cultivars [1,4,17,18]. Among various environmental factors, the thermal regime has emerged as a key variable that appears to explain the differences in FA composition of olive cultivars growing in different environments [4,5,11,18,19]. The effect of thermal regimes appears more pronounced in cultivars with greater phenotypic plasticity, such as ‘Arbequina’ [20,21]. Analyses under field conditions have shown that ‘Arbequina’ oils from warm areas have consistently lower oleic acid (OA) content compared to other cultivars and, consequently, higher linoleic acid (LA) levels [2,4,22]. Temperature-related variations in OA content have also been found in other olive cultivars, such as the Argentine ‘Arauco’ [4], in which manipulative experiments during the oil synthesis period demonstrated that temperatures higher than the seasonal average (5–10 °C or warmer) can consistently decrease OA content [23]. In contrast, the FA composition of other cultivars, such as ‘Coratina’, seems to be relatively stable across environments differing in thermal characteristics [24]. A wide survey of olive cultivars growing on three continents showed considerable phenotypic plasticity in FA composition for many cultivars and a high stability for others [18].

The process that leads to the synthesis of unsaturated fatty acids in olive fruit and determines the OA content of olive oil involves the two key enzymes stearoyl-ACP desaturase (SAD) and oleyl-ACP desaturase (FAD), which are encoded by genes pertaining to the *SAD* and *FAD* families, respectively. To date, many genes have been characterized for each gene family in olive [8,11,13,25]. Based on expression analyses and association studies of *SAD* and *FAD* genes related to fruit FA composition of several olive cultivars, it has been observed that *OeSAD2* and *OeFAD2* genes represent the main contributors to the synthesis of OA and LA, respectively [11,12,13,15,26,27]. The relevance of the *FAD2* genes in the biosynthesis and accumulation of OA and LA has been also widely demonstrated in other crops, such as cotton [28], rapeseed [29], peanuts [30], corn [31,32], yellow mustard [33], oil palm [34] and soybean [35].

Temperature affects fatty acid desaturase genes, either through transcriptional [36] or post-transcriptional [37] regulatory mechanisms. The expression of *FAD2* genes seems to be regulated by temperature and light intensity, whereas that of *FAD7* appears to be affected by high temperatures [9,10]. On the other hand, a number of studies have reported increased transcriptional levels of *SAD* genes in response to low temperatures in different species and plant organs, such as maize [38], rapeseed [39], soybean seeds [40], potato plants [41] and *Ginkgo biloba* leaves [42]. Such increases were associated with the production of higher amounts of unsaturated FA, with differential responses to low and high temperatures, suggesting a fundamental role played by *SAD* genes in the mechanism of cold tolerance. In this regard, Li et al. [43] found that overexpression of *SAD* genes in transgenic potato plants was associated with LA increments in membrane lipids, resulting in improved cold acclimation.

The expansion of olive cultivation to warmer areas than those prevailing in much of the original Mediterranean environment raises the need to deepen our understanding of the possible environmental regulation of factors involved in FA composition at the field level, as well as differential varietal responses. Therefore, the aim of the present study is to determine the effect of the thermal regime of different growing environments during oil synthesis in olive fruits on the expression of main *SAD* and *FAD* genes, as well as the FA composition, in two cultivars, Arbequina and Coratina, over a wide latitudinal gradient in Argentina. From a practical standpoint, the findings obtained in this study could constitute the basis for the planning of new cultivation scenarios, taking into account the thermal records in each area, as well as for the selection of the most suitable genotypes.

## 2. Results

### 2.1. Temperature Regimes during Fruit Growth and Oil Synthesis in the Mesocarp

Environments differ in terms of minimum and maximum temperatures during the period of fruit growth and accumulation of oil in the drupes. All olive orchards from which the samples were collected were irrigated in order to guarantee full water availability.

Figure 1B reports the values of these temperatures at the three sampling times (42, 139 and 173 DAF). At each time point, there was a significant difference among locations. Moreover, the accumulated degree days (ADD) throughout the fruit growth period (starting from flowering up to the date of the last sampling (173 DAF)) showed significant differences between the Catamarca and sites further south, including the nearest La Rioja site (Figure 1C).

### 2.2. Differential Accumulation of Fatty Acids in the Lipids of Fruit Mesocarp According to Cultivar, Environment and Their Relative Thermal Regimes

The four main fatty acids that make up triglycerides of olive oil were considered: palmitic acid (C16: 0), stearic acid (C18: 0), oleic acid (C18: 1) and linoleic acid (C18: 2) (Figure 2).

Fruit samples were collected on three successive dates (42, 139 and 173 DAF); however, it should be noted that 42 DAF, fruits were still in the phase of intense growth, and the oil accumulation in the mesocarp had not yet begun; therefore, the few lipids present during this stage were almost exclusively constitutive membrane lipids and not oil triacylglycerides.

The percentages of palmitic acid ranged between 11% and 31% across all time points and for both cultivars, although in cv. Coratina, contents were almost always slightly lower than those of cv. Arbequina. Significant differences were more pronounced in ‘Arbequina’ between the warmer sites (Catamarca and La Rioja) and that with lower minimum temperatures (NQNa), at least at 42 and 139 DAF, but with an opposite trend, 42 DAF, the lowest values were observed in the warmer environments, whereas 139 DAF, in these environments, the C16:0 showed the highest values. Furthermore, 173 DAF values were very similar among environments and for both cultivars, with cv. Arbequina values higher than those of cv. Coratina.

Stearic acid showed greater variability and higher values 42 DAF, when percentages up to almost 11% were reached and with the lowest values in the coldest sites. Moreover, 139 and 173 DAF percentages were considerably reduced, ranging between 0.5 and 3.5%, with no significant variations for either variety 173 DAF.

As expected, oleic acid reached very high percentages of between approximately 50 and 75% 139 DAF in Arbequina and between 68 and 80% in cv. Coratina, remaining almost unchanged around these values for the variety cv. Coratina 173 DAF, whereas in cv. Arbequina, these values were always lower, at approximately 50% for the hottest environments 139 DAF and up to a minimum of 40 and 47% 173 DAF for the warmer environments of La Rioja and Catamarca, respectively. In colder environments, the values always remained slightly below those of cv. Coratina but still higher than 67%. Furthermore, 42 DAF, the oleic acid percentages were lower, reaching the highest values only for cv. Coratina in the coldest environments of Neuquén. The oleic acid content in fruits 139 and 173 DAF was negatively correlated with the accumulated thermal time (r = −0.89; −0.86, respectively). It should be also stressed that the final oleic acid content in the hottest environments was below the lower limit stated by the International Olive Council (55%) for extra virgin olive oil.

Linoleic acid showed the greatest variability between environments, with values ranging between 5% and 25% 42 DAF for both varieties, with greater values in intermediate environments and the minimum recorded for cv. Coratina in less hot environments. With the start of the oil synthesis process (139 DAF), the values of linoleic acid reduced in all environments, especially for cv. Coratina, whereas for ‘Arbequina’ in warm environments, they ranged between 14% and 20% and were reduced to below 10% in the colder environments. The same trend was observed 173 DAF, with higher values for cv. Arbequina in hot environments, up to almost 30%, whereas for cv. Coratina, the observed differences were not significant, except at the LR and SJa sites, which registered higher values.

Correlation analyses between OA and LA percentages and TT also confirmed this pattern, with negative and positive correlations, respectively, during the last two stages analyzed in ‘Arbequina’; this observation was only evidenced in the intermediate stage in cv. Coratina.

### 2.3. Expression of SAD and FAD Genes in Arbequina and Coratina Cultivars under Different Environments and Fruit Development Time Points

Three genes encoding stearoyl-ACP desaturases (*OeSAD1*, *OeSAD2* and *OeSAD4*) and three encoding oleate desaturases (*OeFAD2-1*, *OeFAD2-2* and *OeFAD6*) previously characterized in other olive cultivars [13] were analyzed in Arbequina and Coratina cultivars in each environment and for the three phenological stages (Figure 3 and Figure 4).

### 2.4. Principal Component Analysis Scores and Loadings

The evolution of *SAD1* gene expression was in line with the expected profile, showing an increase during fruit development, in particular 139 DAF. In fact, in cv. Coratina, the level of expression in warmer areas was higher than that in cv. Arbequina, which always remained very low. Moreover, 173 DAF, differences in cv. Arbequina expression between environments were practically not significant, whereas in cv. Coratina, the expression was generally high in warm environments and low in intermediate and cooler environments.

For both cultivars, *SAD2* was differentially expressed during the first two stages of fruit development. Then, 42 DAF, levels of expression varied by nearly six- and thirteenfold between the hottest and coldest environment, respectively. During intermediate fruit development stages, the lowest expression levels were detected in the coldest environment (Neuquén).

The level of expression of *SAD4* was too low relative to the other *SAD* genes; therefore individuated differences between environments were more limited. Then, 42 DAF, the expression in both cultivars was significantly higher in the warmer environment and intermediate sites, whereas the lowest values were recorded at the southernmost site (NEQb). Furthermore, 139 DAF, there were no differences for cv. Arbequina, and the expression was considerably reduced in cv. Coratina, as well as during the last stage.

With respect to fatty acid desaturase genes, *FAD2-1* expression reached the highest levels 42 DAF in cv. Arbequina in the coldest environments, whereas expression of cv. Coratina remained low, with no differences among sites. Then, 139 DAF, its expression was higher in hot environments only for cv. Coratina, and 173 DAF, significantly higher expression levels were revealed for both cultivars in the hottest environment of Catamarca.

The expression of the *FAD2-2* gene was very high during the first survey 42 DAF, especially in the intermediate and warm environments of San Juan and La Rioja, respectively, and particularly for cv. Coratina, whereas in cv. Arbequina, the records were only significantly higher in the intermediate environments. In the following stages 139 and 173 DAF, the level of expression was considerably reduced compared to the first time point for both cultivars, with significant differences among the environments for both varieties, reaching maximum expression levels in intermediate environments.

The expression of *FAD6* resulted in significant differences among environments only for the Arbequina cultivar at first time point of sampling (42 DAF), although the level of expression was higher for cv. Coratina at all sites, except the coldest site, Neuquén b. Moreover, 139 and 173 DAF, differences among environments were not significant for either cultivar.

The PCA, which included 23 variables, explained 32.80% of total variability for PC1 and 14.89% for PC2 (Figure 5). All environments at 42 DAF were in the positive plot area of PC1 without any correlation with the environment. In contrast, the PCA clearly separated the other two time points in relation to each environment. In fact, all the colder environments for both 139 and 173 DAF were in the plot area where the oleic acid is the principal loading, whereas the warmer areas were located in the area characterized by the maximum gene expression of *SAD1* and *SAD2*, as well as thermal time. The hottest Catamarca and La Rioja environments were separated from the others for the highest values of temperatures (minimum and maximum), accumulated degree days and expression of *SAD1* and partially *SAD2* for both cultivars. Furthermore, a negative correlation was observed between the expression of *FAD2-1* and *FAD2-2* and the oleic acid content (C18: 1), with a positive correlation between *FAD2-2* and linoleic acid content (C18: 2) for both cultivars.

## 3. Discussion

In this study, we analyzed olive fruits at three different time points for their fatty acid contents and gene expression profiles in seven different environments to determine the role of temperature. The selected environments represent the vast temperature variation in Argentina. The obtained results perfectly meet expectations with respect to aspects such as cultivar performance and fruit chemical composition under different thermal regime conditions; moreover, among all studied genes, the expression of *SAD2* followed the same pattern as that of oleic acid, which increased during the study period.

### 3.1. Temperature and Fatty Acid Profiles

The low percentage of oleic acid in hot environments corresponded, as expected, to a parallel increase in the percentage of linoleic acid, confirming that at high temperatures, oleic acid is actively converted into linoleic acid. The oleic acid concentration of cv. Coratina was not affected by the high temperatures and remained constant and higher than that of cv. Arbequina. In all seven studied environments, cv. Arbequina had the highest C18:1 139 DAF, whereas 173 DAF, in the environments with higher maximum and minimum temperatures, the amount of this acid decreased. Only a few cultivars, such as Arbequina and some others used in new intensive groves, are able to achieve consistent yields under new environmental conditions, often reflecting negative changes in their fatty acid profiles [18].

### 3.2. Fatty Acid Profiles and SAD and FAD Gene Expression

The expression profile of *SAD1* increased over time, specially 139 DAF, and remained almost constant by 173 DAF. This gene expression was not affected by temperatures, despite some significant differences 139 DAF in cv. Arbequina and 173 DAF in cv. Coratina.

The pattern of *SAD2* expression was highly related to the quantity of oleic acid in each cultivar. In cv. Coratina, the expression was increased between 42 and 139 DAF, and this pattern was constant 173 DAF, with no significant effect among the warm and cold environments. This was also confirmed by oleic acid synthesis, especially 173 DAF. The Arbequina cultivar had the highest expression 139 DAF, when the amount of oleic acid was at its highest level in all seven environments. Furthermore, 139 DAF and 173 DAF, in the two environments of Catamarca and La Rioja, the level of expression corresponded with oleic acid synthesis; at the first time point, both were at high level, whereas at the later time point, both were decreased. In the cold environments, cv. Arbequina had constant and high expression level 139 and 173 DAF, as well as the quantity of oleic acid.

The results obtained for the *SAD4* gene, the role of which in fatty acid desaturation remains poorly clarified, confirmed that its expression is not relevant to the regulation of fatty acid composition.

The expression of *FAD2-1* and *FAD2-2* was highest during the initial of fruit growth and oil synthesis stages, decreasing during fruit development [13]. The differences among the cultivars and environments were not considerably sufficiently significant to correlate them with linoleic acid synthesis.

The high levels of *FAD6* expression, although not in accordance with the fatty acid composition, confirmed that this gene plays an important role in fatty acid synthesis, although it seems that it is not regulated by temperature.

### 3.3. Overall Comparisons

Clear differences between warm and cold environments were shown by PCA, with temperate environments between them, as well as a clear separation of the late ripening stages (139 and 173 DAF) from the initial ripening stage (42 DAF). Temperatures were found to be determining factors in the expression of *SAD1* and *SAD2* genes and were inversely correlated with the expression of all *FADs*. The expression of both *SAD* genes seems to be relevant for the synthesis of the oleic acid (C18: 1), although *SAD1* appeared to be inversely correlated with stearic acid (C18: 0), as expected.

The expression of *FAD2* genes seems to play a role in shaping the content of oleic acid in the Coratina cultivar. Factors that distinguish samples from the warmer areas of Catamarca and La Rioja from the others include high temperatures and *SAD2* and *SAD1* genes. This seems a contradictory, considering that, theoretically, the greater the expression of *SAD2*, the more oleic acid should be synthesized; however, according to the data obtained, the *SAD2* gene is expressed more in hot environments during the initial stage, generating a considerable amount of oleic acid 139 DAF. Especially in ‘Arbequina’, the oleic acid content drops after this date, probably due to the decreased expression of *SAD2* and the simultaneous upregulation of *FAD2-2* in the last phase.

Considering the analyzed fruit developmental stages, the expression levels of all *SAD* and *FAD* genes tended to be higher in cv. Coratina than in Arbequina. Differences in *SAD* gene expression levels have been reported in other olive cultivars [15,44], as well as that of *OeFAD2-2* [14,45]. Our results confirm that desaturase gene expression during fruit ontogeny may be regulated differently depending on the olive genotype, and regulation seems to take place predominantly at the transcriptional level, as it occurs with desaturates such as Δ^9^ [46,47], Δ^12^ [36,48,49] and Δ^15^ desaturases [50,51] in most plant species.

It was observed that the three *SAD* genes evaluated in all tested growing environments had similar expression patterns in both Arbequina and Coratina cultivars (Appendix A).

In both analyzed cultivars, the *OeFAD2-1* gene seems to be important at the beginning of fruit development, as previously demonstrated in other olive cultivars, such as Leccino [13], Picual [45] and Barnea [26]. Studies by Parvini et al. [14] and Banilas et al. [25] also indicated high expression levels of the *OeFAD6* gene in young fruits. This latter observation does not seem to match with our findings, which show high *OeFAD6* gene expression during more advanced fruit developmental stages (139 and 173 DAF) only in the Italian cultivar Coratina, which should coincide with active oil synthesis.

In summary, the relationships between the accumulated thermal time, the expression of candidate genes *OeSAD2* and *OeFAD2-1* and the content of their synthesis products (oleic and linoleic acids) are more evident in cv. Arbequina than in Coratina. Drupes sampled 173 DAF, when all oil was accumulated, showed higher *SAD2* gene-transcript levels in colder environments, where the highest OA content was recorded. On that sampling date, the colder sites presented with the lowest accumulated TT. On the contrary, both the lowest gene-transcript levels and oleic acid content were found in the hottest environments with the highest TT records.

The results reported in this work based on olive plants grown under field conditions, are inconsistent with those obtained from short-term experiments performed under controlled conditions, which showed slight increases in the transcriptional expression levels of *FAD* genes in Arbequina fruits incubated at 15 °C for 24 h and decreases when fruits were incubated at higher temperatures (35 °C), even if fruits incubated at low or high temperature did not vary significantly in LA contents. The authors attributed the lack of difference in LA contents to the short incubation times [10]. Another study reported increased expression levels of *SAD* genes in the mesocarp of cv. Picual fruits exposed to low temperatures, although this effect was not related to changes in unsaturated fatty acid contents [44].

In order to elucidate how high temperatures may modify olive oil fatty acid composition, Nissim et al. [26] characterized the expression pattern of genes involved in the pathway of fatty acid biosynthesis under high and moderate temperatures, finding that most of the genes were regulated by high temperatures during different stages of fruit development, and many of were cultivar-dependent. The authors suggested *OePDCT* and *OeFAD2* genes as markers for screening of various cultivars to test their tolerance levels to high summer temperatures.

On the other hand, findings from the present study are consistent with previous field studies that suggested the thermal regime as a major factor affecting fatty acid composition of olive oil, all finding that oils from cultivars growing in warm areas had lower oleic acid contents and higher linoleic acid percentages than those from colder environments [5,17,18,24,52]. Correlation analyses testing the accumulated thermal time during the oil synthesis period and the levels of both oleic and linoleic acids indicated different responses to temperature of the olive cultivars. Such responses, in turn, appear to be related to differences in the enzymatic capacities involved in fatty acid desaturation.

## 4. Materials and Methods

### 4.1. Plant Materials, Growing Environments and Experimental Design

Two olive (*Olea europaea* L.) cultivars (Arbequina and Coratina) growing in five different locations in Argentina were evaluated (Figure 1A). The evaluated cultivars were situated in the provinces of Catamarca, La Rioja, San Juan, Mendoza and Neuquén at latitudes ranging from 28° to 38° S. Field studies were carried out during two consecutive crop seasons (2014–2015 and 2015–2016) in San Juan (indicated as a and b) and Neuquén (a, b), whereas the other sites were sampled only in 2015–2016, except Catamarca, which was evaluated in 2014–2015. On the basis of obtaining large differences in the thermal records across the latitudinal gradient, seven growing environments were ultimately selected: (1) Catamarca (CAT), (2) La Rioja (LR), (3) San Juan a (SJa), (4) San Juan b (SJb), (5) Mendoza (MZA), (6) Neuquén a (NQNa) and (7) Neuquén b (NQNb).

In each location, olive trees were selected within an intensively managed commercial orchard. Although rainfall varied between 77 and 475 mm/year, supplemental irrigation was provided to satisfy 100% of crop evapotranspiration (ETc) over the whole growing season in each location. Olive groves where fruits of cvs. Arbequina and Coratina were sampled had a similar age (approximately 8 years old), plant density (approximately 500 trees/ha) and canopy volume (around 12 m^3^/tree). For each cultivar and location, five olive trees were considered, with the three central trees selected for all samplings and measurements and the surrounding two trees as border-guard plants. The trees were chosen based on similar fruit load level (medium–high), which was measured according to the procedure described by the IOC [53]. A new set of trees was used each year.

Three fruit sampling dates, referred to as days after full flowering (DAF), were considered: (I) 42 DAF (fruits before pit hardening), (II) 139 DAF (green–yellow fruit epicarp) and (III) 173 DAF (fruit veraison phase). For each treatment combination (growing environment x sampling date x crop season), 500 g fruits were collected at mid-canopy from the entire tree perimeter. Once collected, fruit samples were immediately frozen in dry ice, transferred to liquid nitrogen within an hour and stored at −80 °C until analysis. Fruits sampled from each selected tree were used to measure *SAD* and *FAD* gene expression and FA composition.

In each location, meteorological data were recorded using an automatic weather station close to each experimental orchard. Temperature data from full flowering to the end of fruit veraison were recorded (Figure 1B). The accumulated thermal time for the same period was calculated (in °C d units) using the single sine, horizontal cutoff method, with critical temperatures of 7 °C (lower limit) and 40 °C (upper limit) (Figure 1C), as suggested by Bodoira et al. [5]. This allowed *SAD* and *FAD* gene expression and FA composition to be assessed as a function of thermal time.

### 4.2. Fatty Acid Analysis

Olive fruits were destoned using a manual olive-pitting machine. The resulting pericarp was submitted to manual removal of the epicarp with the aid of a scalpel. The mesocarp (hereafter referred to as “pulp”) was used for analytical determinations. FA composition was analyzed by direct methylation following the procedure reported by Mousavi et al. [18] with minor modifications. From each fruit sample, a portion of 200 mg of pulp was placed in a reaction tube containing 3.3 mL of methylation solution (methanol:toluene:2.2-dimethoxypropane:sulfuric acid, 39:20:5:2 *V*/*V*) and 1.7 mL of heptane. The tube was heated in a water bath (80 °C) for two hours and cooled to room temperature. A 2 μL aliquot of the resulting supernatant was analyzed by gas chromatography (GC) (Clarus 580, Perkin-Elmer, Shelton, CT, USA) using a fused silica capillary column (30 m × 0.25 mm i.d. × 0.25 μm film thickness) CP Wax 52 CB (Varian, Santa Clara, CA, USA); carrier gas N_2_ at 1 mL/min; split ratio 100:1; column temperature programmed from 180 °C (5 min) to 220 °C at 2 °C/min; injector and detector temperatures at 250 °C, FID. FAME was identified out by comparison of their retention times with those of reference compounds (Sigma-Aldrich, St. Louis, MO, USA) [54].

### 4.3. RNA Extraction

RNA was extracted from fruit pulp samples using Trizol^®^ (Invitrogen) according to the manufacturer’s instructions. To eliminate any DNA contamination, each sample was treated with DNase I (Invitrogen) and then tested by amplifying the glyceraldehyde-3-phosphate dehydrogenase (*OeGAPDH*) as a reference gene [55,56]. Concentration, quality and purity of total RNA were assessed using a Nanodrop ND-1000 spectrophotometer (Thermo Fisher Scientific, Delaware, USA) by checking the absorbances at 230, 260 and 280 nm, as well as the relative ratios of A_260_/A_280_ for protein and of A_260_/A_230_ for salt contamination. A 1.5% agarose gel was run for all extracted samples in order to monitor RNA integrity by controlling the intensity of the double bands and excluding the smearing below them. Single-strand cDNA was synthesized from 500 ng of total RNA using RANDOM primers and SuperScript III Reverse Transcriptase (Thermo Fisher Scientific, Burlington, Canada), as recommended by the supplier. The amplification ability of cDNA was evaluated by PCR amplification of the *OeGAPDH* gene.

### 4.4. Expression Analysis by RT-qPCR

Expression analyses of three *SAD* (*SAD1*, *SAD2* and *SAD4*) and three *FAD* genes (*FAD2-1*, *FAD2-2* and *FAD6*) were performed by quantitative PCR on the reverse-transcribed DNA (RT-qPCR) in a 96-well plate thermocycler StepOne Plus Real-Time PCR System (Thermo Fisher Scientific, Foster City, CA, USA) following the manufacturer’s instructions. Primers for the RT-qPCR experiments were designed using Primer3 version 4.0. Primer efficiency was initially verified by the presence of single PCR product bands after running on agarose gel electrophoresis. Reactions were performed on three biological and two technical replicates for each cDNA sample. Each reaction contained 1 μL of diluted cDNA (1:10), 0.5 μL of each primer (10 pmol/μL) and 6.25 μL of SYBR Green Master Mix reagent (Roche Diagnostics Inc., Basel, Switzerland) in a final volume of 12.5 μL. The following PCR program was used: 1 cycle at 50 °C for 2 min and 95 °C for 10 min; 40 cycles of 95 °C for 15 s and 60 °C for 1 min; and a final cycle of 95 °C for 15 s, 58 °C for 1 min and 95 °C for 15 s. Amplification efficiencies and Ct values were determined for each gene and each tested condition by considering the slope of a linear regression model using LinRegPCR [57]. Only primer pairs that produced the expected amplicons and showed similar PCR efficiency were selected. *OeGAPDH* and elongation factor (*EF 1α*) genes were used as references for sample normalization.

### 4.5. Statistical Analyses

Relative amounts of each transcript were calculated using the 2^−ΔΔCT^ method [58], using the sample with the lowest expression for calibration, which corresponded to the highest level of CT. All molecular and chemical determinations were obtained from triplicate measurements of three independent samples. Statistical differences among treatments were estimated from ANOVA test at the 5% level (*p* ≤ 0.05) of significance for all parameters evaluated. Whenever ANOVA indicated a significant difference, the Di Rienzo, Guzmán and Casanoves (DGC) test was applied to compare the means [59] using InfoStat software (InfoStat version 2020, National University of Córdoba, Córdoba, Argentina).

A principal component analysis (PCA) was applied for a total of 23 variables, and correlation analyses were performed with Pearson’s test. The resulting plots were generated using OriginPro 2022 (OriginLab Corporation, Northampton, MA, USA).

## 5. Conclusions

By exploring FA composition and desaturase gene expression in two olive cultivars (Arbequina and Coratina) grown over a wide latitudinal gradient in Argentina, we observed differential accumulation of oleic and linoleic acids from northern to southern latitudes. Likewise, we identified *OeSAD2* and *OeFAD2-2* as the main genes affecting the concentration of these fatty acids when oil is accumulated. By analyzing the thermal regime of the growing environments, we found that the accumulated thermal time could be a factor affecting the expression of both genes and FA contents. It was also observed that relationships between the accumulated thermal time, the expression of the identified genes and the content of their associated synthesis products (oleic and linoleic acids) were more evident in cv. Arbequina than in ‘Coratina’. This indicates that ‘Arbequina’ FA composition could be more susceptible to temperature than that from ‘Coratina’. Overall, the results suggest a regulatory effect of temperature on the expression of the abovementioned desaturase genes, which appears to depend on the olive cultivar. At a more basic level, these findings deepen our understanding of the possible environmental regulation of factors involved in olive oil FA synthesis. From a practical standpoint, the reported results could serve as a basis for further studies evaluating growing environments or conditions for implantation of new olive orchards and for the selection of better-adapted genotypes.

## Figures and Tables

**Figure 1 plants-12-00054-f001:**
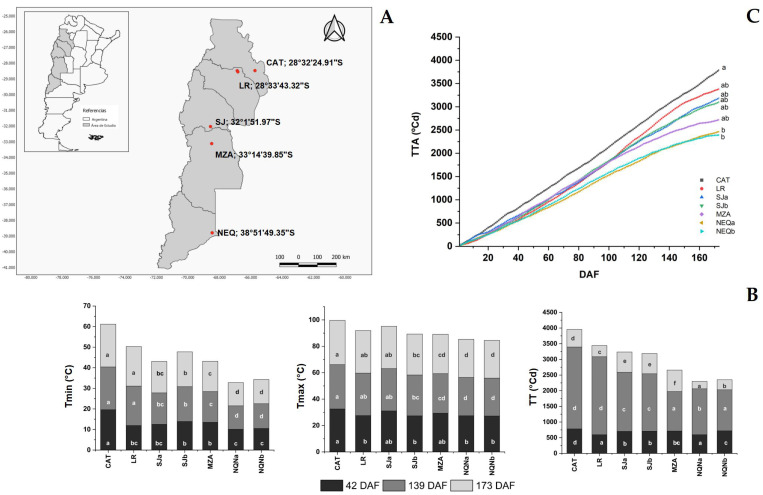
(**A**) Map of the study area in Argentina covering multiple provinces along the latitudinal growth gradient (Catamarca (CAT) 2014/15, La Rioja (LR), 2015/16, San Juan (SJa), 2014/15, San Juan (SJb), 2015/16, Mendoza (MZA), 2015/16, Neuquén (NQNa), 2014/15 and Neuquén (NQNb), 2015/16). The geographical location of the sampled sites is indicated in red. (**B**) Average minimum and maximum temperatures and accumulated degree days detected at 42, 139 and 173 DAF for the different environments; (**C**) accumulated thermal time (TT, °C d) registered from full flowering to fruit veraison in the seven analyzed growing environments. Different letters correspond to significant differences at *p* < 0.05 among environments for a given phenological stage.

**Figure 2 plants-12-00054-f002:**
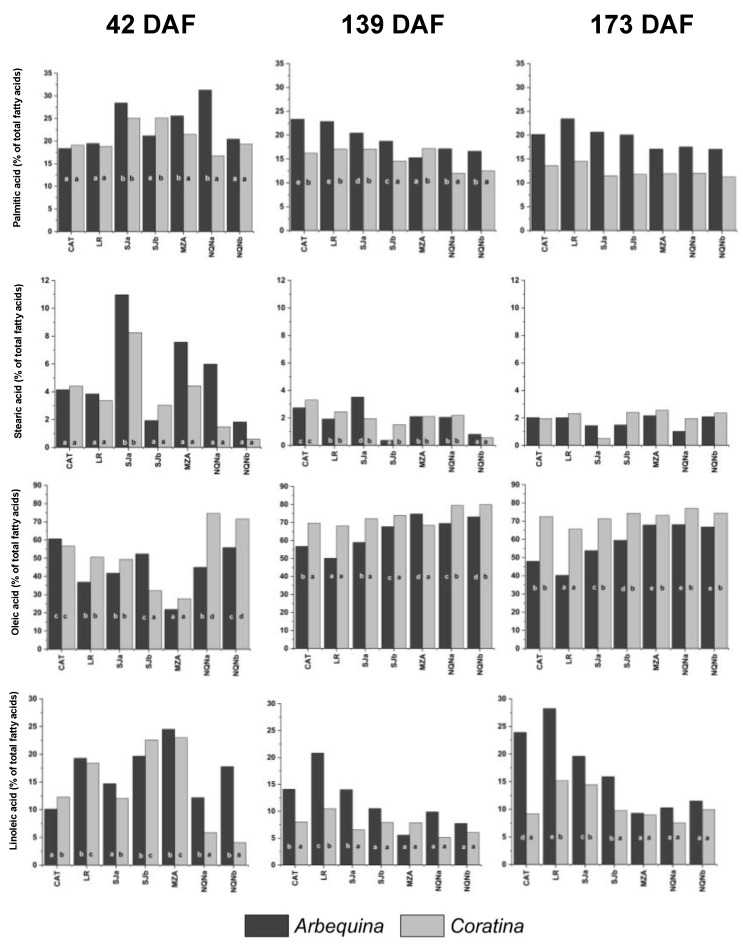
Percentage of main fatty acids (palmitic, stearic, oleic and linoleic acids) during three fruit phenological stages (42, 139 and 173 days after full flowering (DAF)) in cvs. Arbequina and Coratina in the seven analyzed growing environments (Catamarca (CAT), 2014/15; La Rioja (LR), 2015/16; San Juan (SJa), 2014/15; San Juan (SJb), 2015/16; Mendoza (MZA) 2015/16; Neuquén (NQNa) 2014/15; and Neuquén (NQNb), 2015/16). Different letters correspond to significant differences at *p* < 0.05 among environments for a given phenological stage.

**Figure 3 plants-12-00054-f003:**
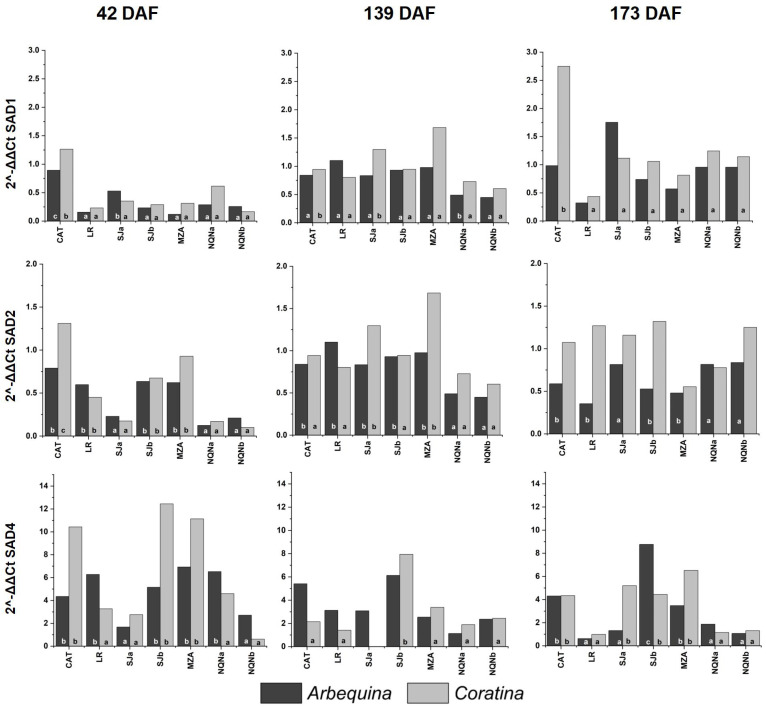
Expression of the *OeSAD* gene family during three fruit phenological stages (42, 139 and 173 days after full flowering (DAF)) in cvs. Arbequina and Coratina in the seven analyzed growing environments analyze (Catamarca (CAT), 2014/15, La Rioja (LR), 2015/16, San Juan (SJa), 2014/15; San Juan (SJb), 2015/16; Mendoza (MZA), 2015/16; Neuquén (NQNa), 2014/15; and Neuquén (NQNb), 2015/16). Different letters correspond to significant differences at *p* < 0.05 among environments for a given phenological stage.

**Figure 4 plants-12-00054-f004:**
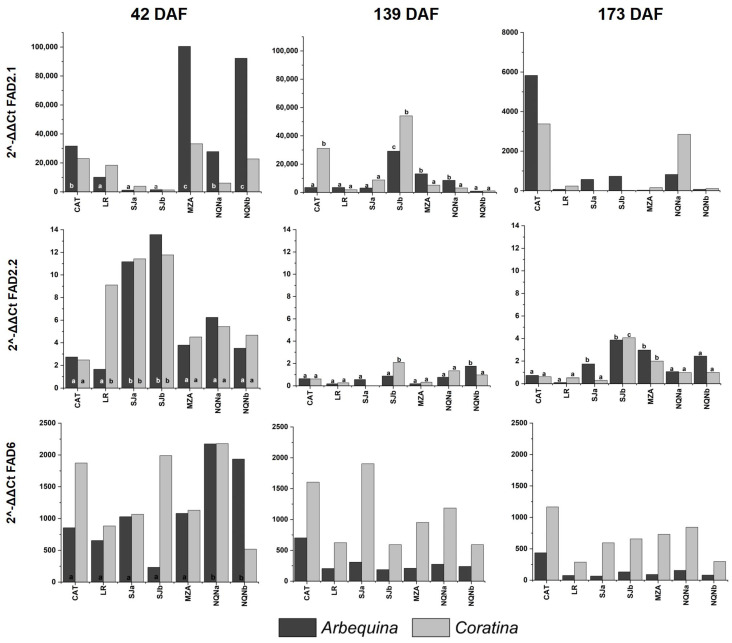
Expression of the *OeFAD* gene family during three fruit phenological stages (42, 139 and 173 days after full flowering (DAF)) in cvs. Arbequina and Coratina in the seven analyzed growing environments (Catamarca (CAT) 2014/15; La Rioja (LR), 2015/16; San Juan (SJa), 2014/15; San Juan (SJb), 2015/16; Mendoza (MZA), 2015/16; Neuquén (NQNa), 2014/15; and Neuquén (NQNb), 2015/16). Different letters correspond to significant differences at *p* < 0.05 among environments for a given phenological stage.

**Figure 5 plants-12-00054-f005:**
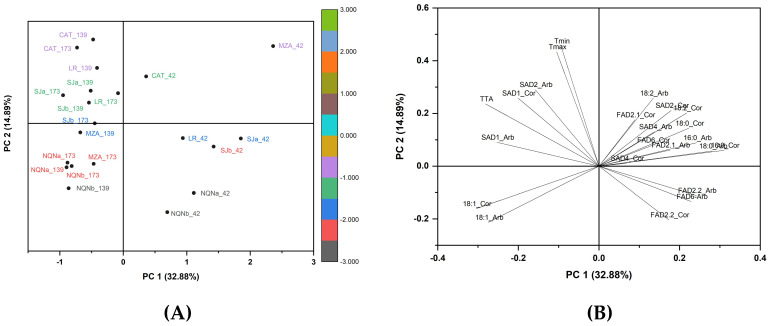
(**A**) Score plot of principal components 1 and 2 for the seven growing environments analyzed in cvs. Arbequina and Coratina; (**B**) loading plot for chemical, molecular and thermal regimes considered in both olive cultivars.

## Data Availability

Not applicable.

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
