# Peer review of "How Temperatures May Affect the Synthesis of Fatty Acids during Olive Fruit Ripening: Genes at Work in the Field"

_plants, 2022, doi:10.3390/plants12010054_

Round 1

Reviewer 1 Report

This manuscrip looks for the differential impact of temperatures on the fatty acids composition of two olive tree cultivars.

It is a complete research, two points to be checked, first is english which is quite correct but some sentences should be reviewed. And the form of the figures which deserve a little the work as we cannot see them properly, too small.

See more detailed comments on minor points on the document attached

Author Response

Dear Reviewer

We appreciate your suggestions, as they allowed us to substantially improve the article. We have incorporated all modifications in the new version of the manuscript. We have also modified the format of all Figures to facilitate their visualization.

Thank you very much

Best regards

Reviewer 2 Report

In the submitted manuscript the authors described the effects of the thermal regime on the production of unsaturated fatty acids during oil synthesis in the drupes. In particular, they evaluated the levels of four fatty acids and the expression levels of six genes involved in fatty acids synthesis in two olive cultivars sampled in 7 different environments.

One aspect is not clear to me. Why did the authors decide to consider two growing seasons only for two sampling sites? Doing it for all the sites would have made more sense to me.

Overall, the topic is interesting and the results are convincing and well argued in the Discussion section. 

I have only a few minor comments that may be addressed:

1.     Please, check all figures. Some of them are very hard to read and the letters related to the statistical difference are missed sometimes.

2.     I believe that in Figure 1 the representation of maximum and minimum temperature and accumulated thermal time would more clear through a table instead of histograms. 

3.     Line 279: do you refer to cv. Coratina instead of Arbequina?

Author Response

Dear Reviewer,

We are thankful for all the suggestions to improve our article. Next, we clarify your queries:

One aspect is not clear to me. Why did the authors decide to consider two growing seasons only for two sampling sites? Doing it for all the sites would have made more sense to me.

To explain this point, in the section 2.1 (Materials and Methods), we have added the following paragraph: On the basis of obtaining large differences in the thermal records across the latitudinal gradient, seven growing environments were finally selected: 1) Catamarca (CAT); 2) La Rioja (LR); 3) San Juan a (SJa); 4) San Juan b (SJb); 5) Mendoza (MZA); 6) Neuquén a (NQNa); 7) Neuquén b (NQNb).

  1. Please, check all figures. Some of them are very hard to read and the letters related to the statistical difference are missed sometimes.

We have already modified them.

  1. I believe that in Figure 1 the representation of maximum and minimum temperature and accumulated thermal time would more clear through a table instead of histograms. 

We consider that the Figures are more illustrative to show the environmental data, and thus visualize the differences over the wide latitudinal gradient of olive growing in our country.

  1. Line 279: do you refer to cv. Coratina instead of Arbequina?.

We have modified this paragraph as follows: For both cultivars, SAD2 resulted differentially expressed at the first two stages of fruit development. At 42 DAF, levels of expression varied nearly to six and thirteen folds, respectively, between the hottest and the coldest environment. At intermediate fruit development stages, the lowest expression levels were detected in the coldest environment (Neuquén).

Thanks a lot

Best regards